# Photoactivatable metabolic warheads enable precise and safe ablation of target cells in vivo

Sam Benson[1], Fabio de Moliner[1], Antonio Fernandez[1], Erkin Kuru[2,3], Nicholas L. Asiimwe [4], Jun-Seok Lee [5], Lloyd Hamilton[6], Dirk Sieger [6], Isabel R. Bravo[1], Abigail M. Elliot[1], Yi Feng [1✉] & Marc Vendrell [1✉]

Photoactivatable molecules enable ablation of malignant cells under the control of light, yet current agents can be ineffective at early stages of disease when target cells are similar to healthy surrounding tissues. In this work, we describe a chemical platform based on amino-substituted benzoselenadiazoles to build photoactivatable probes that mimic native metabolites as indicators of disease onset and progression. Through a series of synthetic derivatives, we have identified the key chemical groups in the benzoselenadiazole scaffold responsible for its photodynamic activity, and subsequently designed photosensitive metabolic warheads to target cells associated with various diseases, including bacterial infections and cancer. We demonstrate that versatile benzoselenadiazole metabolites can selectively kill pathogenic cells - but not healthy cells - with high precision after exposure to non-toxic visible light, reducing any potential side effects in vivo. This chemical platform provides powerful tools to exploit cellular metabolic signatures for safer therapeutic and surgical approaches.

[1] Centre for Inflammation Research, Queen's Medical Research Institute, The University of Edinburgh, Edinburgh, UK. [2] Department of Genetics, Harvard Medical School, Boston, MA, USA. [3] Wyss Institute for Biologically Inspired Engineering, Boston, MA, USA. [4] Molecular Recognition Research Center, Korea Institute of Science and Technology (KIST) & Bio-Med Program KIST-School UST, Seoul, South Korea. [5] Department of Pharmacology, Korea University College of Medicine, Seoul, South Korea. [6] The Centre for Discovery Brain Sciences, The University of Edinburgh, Edinburgh, UK. ✉email: yi.feng@ed.ac.uk; marc.vendrell@ed.ac.uk

Since the early 1990s, photodynamic therapy (PDT) has been clinically used as a therapeutic approach to ablate pathogenic and malignant cells in various diseases, from bacterial infections to cancer[1]. PDT combines light, oxygen, and photosensitizers to produce toxic reactive oxygen species locally, and cause cell death. A broad range of chemical architectures have been reported as effective PDT agents, including organic and inorganic molecules[2–6], superoxide radical generators[7, 8], core-shell metal nanoparticles[9, 10], metal-organic frameworks[11–13], and mesoporous organosilica nanoparticles[14]. Recently, the groups of Gasser, Chao, and Sadler described the use of highly oxidative iridium photocatalysts to induce intracellular redox imbalance and kill cancer cells even in the presence of low concentrations of oxygen[15]. In addition to exhibiting high phototoxic potential, a critical requirement for PDT agents is to exhibit toxicity only in target cells so that therapeutic benefit can be achieved with tolerable light doses and minimal side effects in healthy tissues.

Different strategies have been described to fine-tune the structures of photosensitizers and enhance their cell selectivity. Activatable agents exploit some of the biochemical features of tumors (e.g., low pH, glutathione, proteases) to release photosensitizers in cancerous tissues, showing promise in reducing bystander toxicity[10, 16–20]. Alternatively, photosensitizers can be conjugated to antibodies, which favor accumulation in tissues that express specific cell-surface markers[21, 22]. Whereas these approaches can be effective in the presence of abundant biomarkers, they show limitations at early stages of disease when the biochemical and proteomic signatures of target cells are not dissimilar to those of surrounding healthy tissues (e.g., microtumors). Chemical designs to directly deliver photosensitizers to individual pathogenic and malignant cells under in vivo physiological conditions would significantly extend the clinical applications of PDT.

Cellular metabolism is a reliable indicator of disease onset and frequently used in the clinic (e.g., metabolomics can assist in the detection and therapy stratification of infection[23] and cancer patients)[24, 25]. The metabolic patterns of cells belonging to different taxa (e.g., prokaryotic vs eukaryotic cells) or with altered energetic demands (e.g., cancer vs healthy cells, due to the Warburg effect)[26] are unique; however, light-controlled ablation of cells on the basis of differential or altered metabolism has been negated by the lack of photosensitive chemical structures that retain the properties of native metabolites.

Most photosensitizers are derived from bulky fluorescent scaffolds[27–29], which impair their active uptake as photosensitive metabolites. Smaller fluorophores, such as pyrroloindolizines, nitrobenzodiazoles, and naphthylamines[30–32] are more appropriate to retain metabolite function. In this work, we have designed a platform based on amino-substituted benzoselenadiazoles as the smallest photosensitizers reported to date and used them to build photodynamic agents that recapitulate the uptake of native metabolites, unlike some commercial photosensitizers. Benzoselenadiazole probes can selectively kill cells exhibiting altered metabolism with high spatial resolution and only after exposure to nontoxic light. Importantly, we demonstrate that benzoselenadiazoles are compatible with different metabolite structures (e.g., amino acids, saccharides) and can be used to ablate a variety of disease-causing cells, from bacterial pathogens in vitro to glioblastoma microtumors in vivo. This chemical platform demonstrates how small photosensitizers can harness early metabolic signatures to safely eliminate harmful cells in vivo without damaging healthy tissues.

## Results
### Amino-substituted nitrobenzoselenadiazoles are small photosensitizers. In order to generate photosensitizers that would not impair the biomolecular properties of native metabolites, we

focused on the nitrobenzodiazole core as an example of a small and uncharged chromophoric scaffold[33]. With the exception of nitrobenzoxadiazoles (e.g., **NBD-Cl**, Fig. 1a), which are commercially available, we employed the diamine **1** to prepare nitrobenzodiazoles including different bridging atoms, namely sulfur, selenium, or carbon (Fig. 1a, full synthetic and characterization details in Supplementary Information). Thio- and selenoderivatives (**3**, **4**) were prepared by condensation of the diamine **1** with *N*-sulfinylaniline or $SeO_2$ respectively, followed by nitration of the position 7 under acidic conditions. The synthetic route for the carbon derivative **5** was modified due to the lability of aminoacetals under strong acidic conditions, which would deter the nitration step. Therefore, we treated nitrobenzoselenadiazole **4F** with HI and HCl to remove the selenium atom and performed a Cu-catalyzed condensation with acetone to obtain the corresponding carbon-bridged nitrobenzodiazoles (**5**). All nitrobenzodiazoles were treated with diethylamine and isolated as the corresponding aminoderivatives (**2a–5a**, Fig. 1a), which absorb in the visible spectral range. Despite some of these compounds having a relatively high number of N atoms, all of them fulfill the rule of 3 (i.e., the number of C and O atoms divided by the number of N atoms is equal or larger than 3) and therefore are deemed safe to handle and store[34].

Compounds **2a–5a** showed absorbance maxima wavelengths between 480 and 565 nm (Supplementary Table 1); however, only the selenium derivative (**4a**) displayed high cellular toxicity after exposure to the light dose generally used in clinical PDT (i.e., 37J cm$^{-2}$, Fig. 1b)[35]. Remarkably, the nitrobenzoselenadiazole **4a** displayed minimal cell death in the dark (<10%) and almost complete cell ablation (>90%) after irradiation, whereas none of the other compounds (**2a**, **3a**, and **5a**) showed significant differences between dark and light-induced cytotoxicity (Fig. 1b). These observations were supported by DFT (Density Functional Theory) calculations (Supplementary Fig. 1). These indicated that, whilst all compounds had sufficient energy to excite $^3O_2$ to $^1O_2$ via their S0-T2 transition, the energy levels of the benzoselenadiazole **4a** showed the smallest energy gap for intersystem crossing between S$^1$ and T$^2$ excited states, which favors the occupation of the T$^2$ state and the probability of singlet oxygen production under light irradiation. Furthermore, the degree of spin orbital coupling induced by selenium is 5-fold higher than for other heteroatoms[36], which significantly increases the probability of accomplishing this formally forbidden electronic transition. Collectively, these results highlight the heavy selenium atom as a critical feature to endow the nitrobenzodiazole core with unique photosensitive capabilities.

Previous reports have described the importance of the substituents in the position 4 of nitrobenzodiazoles to fine-tune their optical properties[37]. Therefore, we synthesized a subset of nitrobenzoselenadiazoles where the position 4 was functionalized with different nucleophiles and aromatic groups to assess whether those would influence phototoxicity. Benzoselenadiazoles including amines, phenols, thiophenols, and aryl groups (compounds **6–9**, Fig. 1c) were synthesized by nucleophilic substitution under basic conditions or by conventional Pd-catalyzed Suzuki couplings (full synthetic and characterization details in Supplementary Information). Spectral characterization confirmed that amino-substituted analogs presented an effective push-pull dipole as indicated by their longer excitation wavelengths (Fig. 1d). Amino-substituted nitrobenzoselenadiazoles showed the highest phototoxicity in cells compared to all other substituents and to the control compound **4F** (Fig. 1d). We also examined whether the incorporation of additional heavier atoms in nitrobenzoselenadiazoles—such as iodine, already reported for the successful conversion of other fluorescent structures into photosensitizers[38]—would further

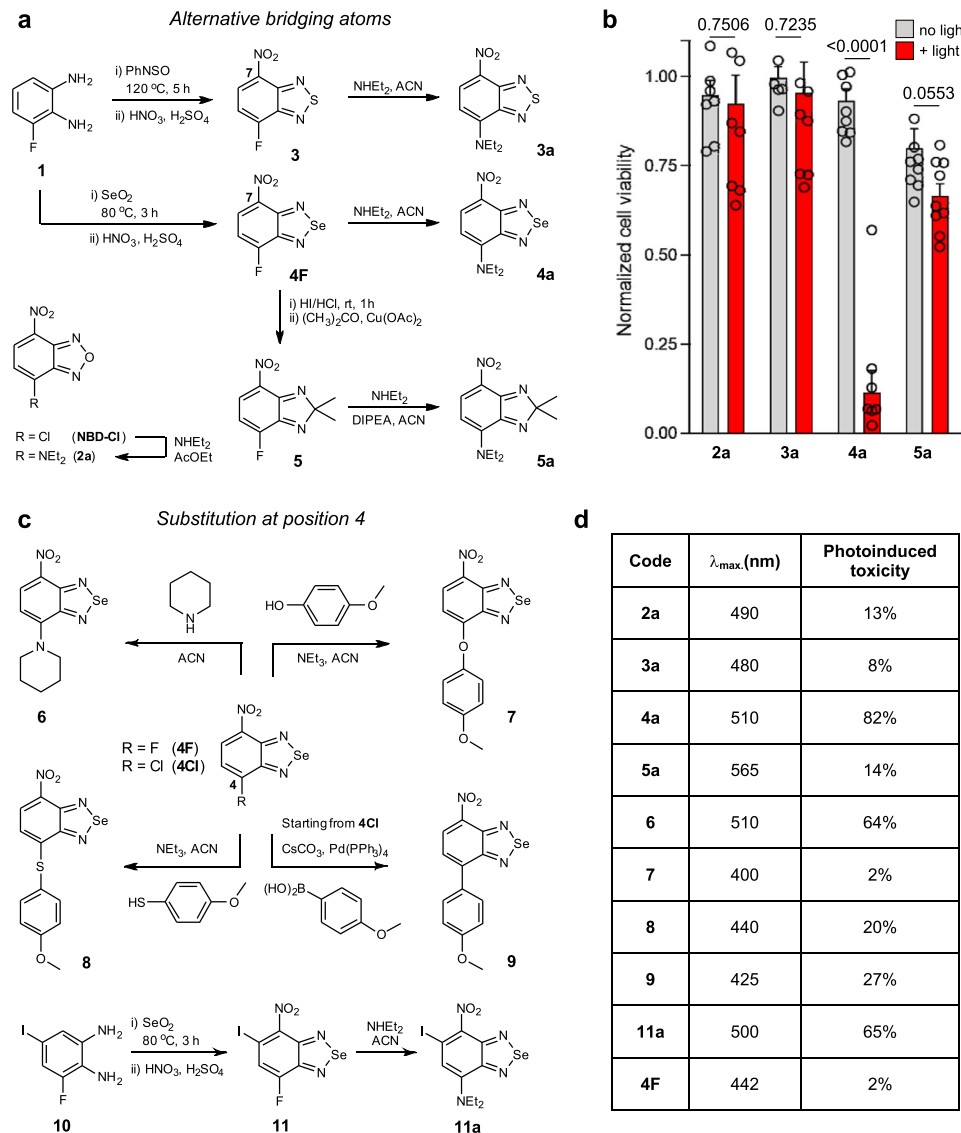

**Fig. 1 Synthetic routes for the preparation of small photosensitizers. a** Condensation of **1** with PhNSO or SeO$_2$ followed by nitration led to compounds **3** and **4F**, respectively. Reduction of **4F** followed by Cu-catalyzed addition of acetone rendered compound **5**. Nucleophilic substitutions of **2–5** with diethylamine yielded **2a–5a**. **b** Human U87 cells were incubated with **2a–5a** (100 μM) in Krebs–Ringer buffer and illuminated (red bars) or not (gray bars) with a ThorLabs M530L3 LED (10 mW, 37 J cm$^{-2}$, red). Cell viability was assessed 16 h post illumination using a TACSR MTT cell proliferation assay with values normalized to the viability of untreated cells. Data presented as mean values ± SEM (n = 3 independent experiments). **c** Nucleophilic substitution of **4F** with piperidine, 4-methoxyphenol, and 4-mehtoxythiophenol led to compounds **6**, **7**, and **8**, respectively. Suzuki coupling of **4Cl** with methoxybenzene-4-boronic acid rendered compound **9**. Condensation of compound **10** with SeO$_2$ followed by nitration and nucleophilic substitution resulted in compound **11a**. **d** Maximum excitation wavelengths and phototoxicity of compounds **2a–11a** and **4F** in human cells as determined in (**b**). Values indicate the decrease in viability between irradiated and non-irradiated cells after incubation with compounds **2a–11a** (100 μM). P values were obtained from two-tailed unpaired t tests. Source data are available.

improve their phototoxic potential. We employed the commercially available mono-iodinated diamine **10** to synthesize 9-diethylamino-6-iodo-7-nitrobenzoselenadiazole (**11a**) as described above (Fig. 1c; full synthetic details in Supplementary Information) and compared its behavior to the parent non-iodinated compound (**4a**). Both compounds **4a** and **11a** presented similar excitation wavelengths around 500 nm, with the latter exhibiting dark cytotoxicity and the non-iodinated derivative showing higher ratios of phototoxicity, making it more suitable for the preparation of photosensitive metabolites. Finally, we examined the mechanism by which amino-substituted nitrobenzoselenadiazoles induce oxidative death[39] and confirmed that, upon light irradiation, they predominantly produce singlet oxygen over radical anions (approximately in a 75:25 ratio,

Supplementary Fig. 2). Amino-substituted nitrobenzoselenadiazoles are therefore one of the smallest photosensitizer structures reported to date.

**Benzoselenadiazole-conjugated amino acids retain stereospecific recognition of bacterial cells.** In view of the suitable properties of benzoselenadiazoles as small and uncharged photosensitizers, we examined their impact on the molecular recognition properties of metabolites, such as amino acids. VanNieuwenhze, Brun and co-workers have demonstrated that derivatives of D-amino acids (e.g., D-alanine) but not L-amino acids can be recognized by the metabolic cell wall machinery of bacterial cells and incorporated into

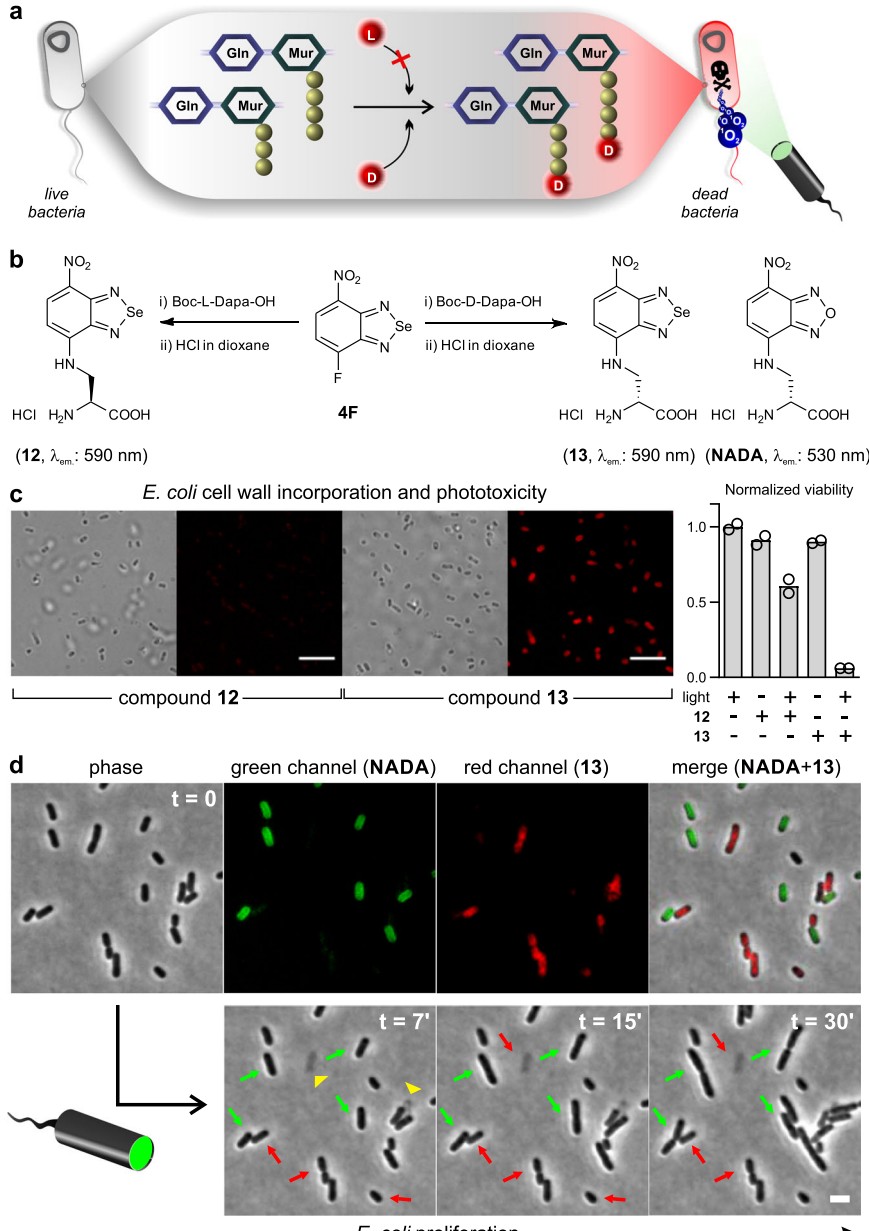

**Fig. 2 Benzoselenadiazole D-amino acids can be incorporated into bacteria for light-controlled antimicrobial ablation. a** ᴅ-alanine is incorporated into the positions 4 or 5 of peptidoglycan structures. Incorporation followed by illumination results in the production of singlet oxygen and concomitant cell death (Gln: *N*-acetylglucosamine, Mur: *N*-acetylmuramic acid). **b** Synthesis of the amino acids **12** and **13** by conjugation of **4F** with enantiomers of 2,3-diaminoproprionic acid. **c** Brightfield and fluorescence microscopy images of *E. coli* after incubation with compounds **12** and **13** (250 µM). Scale bar: 10 µm. The viability of bacterial cells was measured by optical density (600 nm) after incubation at 30 °C with or without compounds **12** or **13** and illumination using white LED. Data normalized to the viability of light-only treated cells and presented as mean values ± SEM (*n* = 2 independent experiments). **d** Time-lapse monitoring of *E. coli* that had been pretreated with **NADA** or compound **13** (both 100 µM). Representative brightfield and fluorescence microscope images (from three independent experiments) with excitation at 470 nm (196 mW) and 508 nm (62 mW) were taken every minute for 30 min. Green and red arrows indicate **NADA**- and **13**-labeled bacteria, respectively; yellow arrowheads indicate exploding **13**-labeled bacterial cells after treatment. Scale bar: 2 µm. Source data are available.

their peptidoglycan cell walls (Fig. 2a)[40, 41]. Therefore, we prepared benzoselenadiazole derivatives of ᴅ- and ʟ-alanine to examine whether the stereospecific incorporation of the ᴅ-enantiomer would be retained after conjugation to the photosensitizer. Two amino acids (**12**, the ʟ-alanine surrogate and **13**, the ᴅ-alanine surrogate, Fig. 2b) were synthesized by nucleophilic substitution of the fluorinated benzoselenadiazole **4F** with the two enantiomers of 2,3-diaminopropionic acid (Dapa) (Fig. 2b, full synthetic details in Supplementary Information). Both amino acids exhibited similar

optical properties and singlet oxygen quantum yields, indicating comparable phototoxicity potential (Supplementary Table 1).

During the synthesis of peptidoglycans in the bacterial cell wall, the nascent peptide chains are specifically capped with ᴅ-alanines, which can be selectively exchanged by exogenously provided unnatural ᴅ-amino acids (Fig. 2a)[42]. To monitor the effect of the amino acids in bacterial cells, we grew Gram-negative *Escherichia coli* in the presence of compounds **12** or **13** and imaged the cells shortly after by fluorescence microscopy. We observed that *E. coli*

cells incubated with the D-amino acid **13** -but not the L-amino acid **12**- showed uniform labeling of the bacterial septa, indicating that the benzoselenadiazole scaffold retained the selectivity of D-alanine (Fig. 2c). We confirmed by mass spectrometry that the D-amino acid **13** was covalently incorporated into the peptidoglycan cell wall (Supplementary Fig. 3) and that the incorporation was independent of the bacterial strain, as shown in experiments with Gram-positive bacteria, such as the model species *Bacillus subtillis* and the pathogen *Staphylococcus aureus* (Supplementary Fig. 3). We also examined whether compound **13** could kill *E. coli* after light irradiation. *E. coli* bacterial colonies were treated with compounds **12** (as a negative control) or **13** and kept in the dark or irradiated with light, followed by proliferation measurements under both conditions. As shown in Fig. 2c, high levels of cell death were observed in **13**-treated and light-irradiated *E. coli* cells (<10% cell viability) whereas minimal toxicity was detected in non-irradiated cells or cells that had been incubated with the control compound **12**. Finally, we performed time-lapse imaging of *E. coli* cocultures that had been separately pre-labeled with compound **13** or **NADA** (Fig. 2b). **NADA** is a reported nitrobenzoxadiazole D-alanine analog lacking the selenium atom and able to fluorescently stain bacterial cell walls[43]. The amino acids **NADA** and **13** fluorescently labeled *E. coli* cells ($\lambda_{em}$ (**NADA**): 530 nm, $\lambda_{em}$ (**13**): 590 nm), and their growth was monitored for 30 min while irradiating the cells at regular intervals. Remarkably, most **NADA**-labeled cells showed clear signs of continued growth and proliferation over time, whereas cells that had been pretreated with compound **13** appeared as lysed or showed arrested cell growth (Fig. 2d). These results support that the local production of singlet oxygen after the irradiation of peptidoglycan-incorporated benzoselenadiazoles can selectively target and kill bacterial cells with high spatial resolution and without affecting neighboring cells in the close vicinity. Given that compound **13** can be effectively incorporated into Gram-negative and Gram-positive bacteria and cause cell death upon light irradiation, these molecules could open avenues to develop antibiotic regimes overcoming complications associated to antimicrobial resistance in pathogenic bacteria.

**A 2-deoxyglucose benzoselenadiazole photosensitizer enables light-controlled ablation of human glioblastoma cells in vitro and in 3D spheroids.** Glioblastomas are the most common and one of the most aggressive types of brain cancer, where surgical treatments require highly precise and safe strategies to remove malignant cells without affecting healthy tissues. Brain tumor cells are very glycolytic and show increased levels of glucose consumption when compared to healthy tissue[44], hence we decided to assess whether glucose-derivatized benzoselenadiazoles could kill human glioblastoma cells after exposure to nontoxic light.

Two D-glucose analogs (compounds **14** and **15**, Fig. 3a) were synthesized by condensation of the benzoselenadiazole scaffold **4F** with two 2-D-deoxyglucosamine analogs. Both compounds were isolated as the mixture of α- and β- anomers of D-glucose (characterization data in Supplementary Information). Compound **15** showed better water solubility and photoreactivity than compound **14**, thus we assessed whether it could be actively taken up by GLUT1/3 transporters in glioblastoma cells, the most abundant glucose transporters in human U87 cells[45]. As controls, we also prepared D-glucosamine derivatives of previously reported photosensitizers, such as Nile Blue reported by Fowley and co-workers[46] and tetraphenylethylene (TPE) reported by the Bin Liu et al. for metabolic PDT[47] (compounds **16** and **17**, respectively, Fig. 3b). First, we compared the uptake for all D-glucose derivatives in U87 cells. As shown in Fig. 3c, compound **15** — but not Nile

Blue (**16**) or TPE (**17**) derivatives—exhibited reduced uptake in cells that had been pre-incubated with the GLUT inhibitor cytochalasin B, suggesting that compound **15**—unlike other photosensitizers—can enter glioblastoma cancer cells through active uptake via GLUT transporters. We also analyzed whether the phototoxicity of compound **15** in U87 cells was dependent on GLUT-mediated transport and observed that its activity was significantly blocked with cytochalasin B (Fig. 3c). Furthermore, compound **15** showed the highest singlet oxygen generation quantum yield among all aminobenzoselenadiazoles (Fig. 3d), with values around 25%—comparable to TPE—and produced effective amounts of singlet oxygen by single-photon (520 nm) in vitro and in cells as well as by two-photon (970 nm) excitation (Supplementary Fig. 4). Given the technical complexity of two-photon in vivo experiments and the limited availability of multi-photon illumination in clinical settings, we decided to perform our biological characterization using single-photon excitation.

We cultured human U87 glioblastoma cells and incubated them with compound **15** to directly image the viability of the cells in the dark as well as after light irradiation. Because photosensitizers producing singlet oxygen induce oxidative cell death via apoptosis[48], we used the apoptotic marker Annexin V as an indicator of the toxicity caused by compound **15**. Confocal fluorescence microscopy confirmed that U87 glioblastoma cells that had taken up compound **15** (as shown by intracellular red staining, Fig. 4a) were apoptotic (blue cell membrane staining by Annexin V-Pacific Blue, Fig. 4a) only after light irradiation but not in the dark. On the other hand, single treatments (i.e., compound **15** only or light irradiation only) did not cause cell death as confirmed by fluorescence microscopy and quantitative flow cytometry (Fig. 4b). We also confirmed that compound **15** would compete with unlabeled D-glucose, which drastically reduced the uptake and phototoxicity of compound **15** (Fig. 4a). Furthermore, as a control we used **2-NBDG**, the non-photosensitive nitrobenzoxadiazole glucose analog (Fig. 3a) and confirmed that the selenium atom in the benzoselenadiazole core is essential to cause light-induced oxidative death (Fig. 4b). Finally, we examined the toxicity of compound **15** in 3D cultures from U87-nlsCrimson glioblastoma cells, which express the fluorescent protein E2Crimson in their nuclei as a cell viability marker ($\lambda_{exc/em}$: 610/645 nm). U87-nlsCrimson cells were cultured in ultra-low adherence plates at 37 °C, forming spheroids of around 250 μm in diameter after 3 days. We compared the size of the spheroids in the absence and presence of compound **15** as well as before and after irradiation (Fig. 4c, d). Compound **15**-treated and irradiated spheroids showed a significant decrease in the size of the cell viable core when compared to any of the other incubation conditions, indicating that compound **15** can effectively kill cells in multiple layers of 3D glioblastoma spheroids.

**Compound 15 shows high potency and selectivity for glioblastoma cells in cocultures of human brain cells.** In view of the potential of compound **15** to act as a PDT agent for glioblastoma, we examined its ability to selectively kill glioblastoma cells when they were cultured in the presence of non-cancerous brain cells. We cocultured the glioblastoma cell line U87-nlsCrimson with human HCN-2 cells as representative non-cancerous neuronal precursor cells[49]. HCN-2 cells were pre-stained with CellMask Green and cocultured with U87-nlsCrimson cells (Fig. 5a). Under normal coculture conditions, both HCN-2 and U87 cells were viable, as indicated by fluorescence emission of their respective markers (i.e., CellMask Green for HCN-2, E2Crimson for U87, Fig. 5b). Cocultures were then incubated with compound **15**, irradiated with visible light at 37 J cm$^{-2}$, and monitored by fluorescence confocal microscopy. Notably, the combination

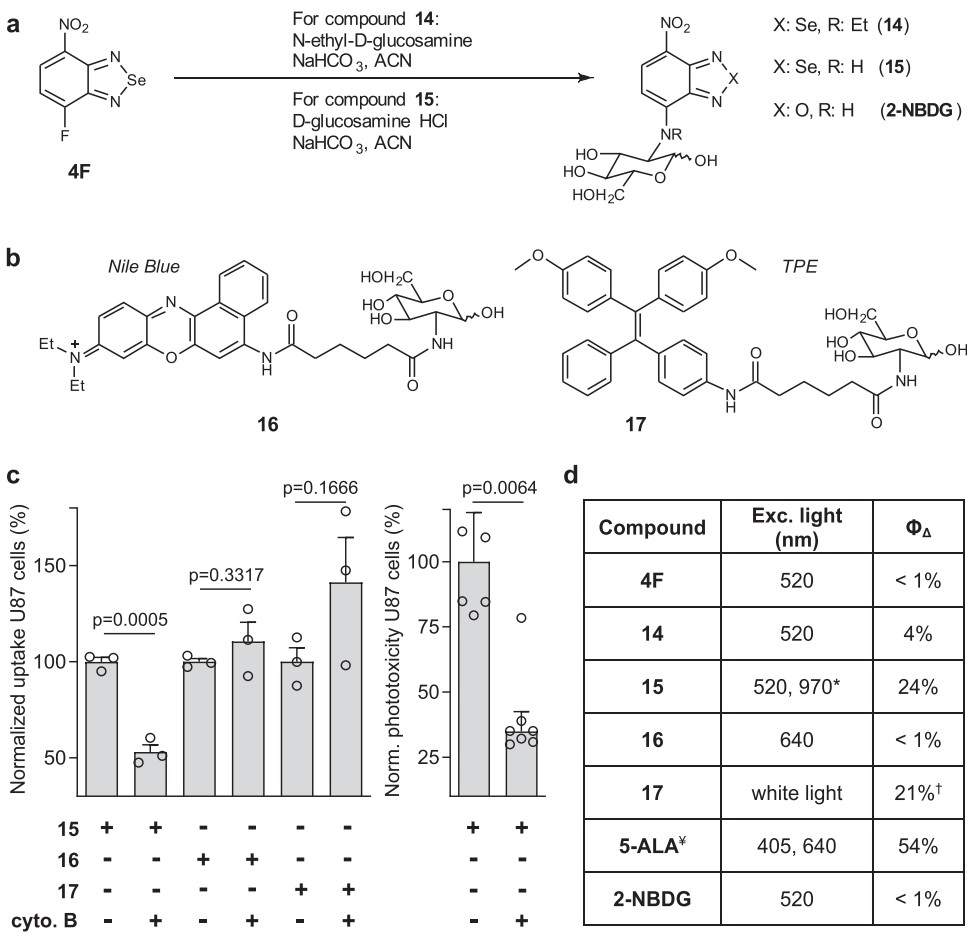

**Fig. 3 D-Glucose derivatives of benzoselenadiazole — but not of other photosensitizers—are recognized by GLUT transporters. a** Chemical synthesis of the D-glucose aminobenzodiazole compounds **14** and **15**, and the non-photosensitive control compound **2-NBDG**. **b** Structures of D-glucose derivatives of the photosensitizers Nile Blue (compound **16**) and tetraphenylethylene (TPE, compound **17**). Full synthetic and characterization details described in the Supplementary Information. **c** Flow cytometric quantification (gating: Supplementary Fig. 5) of uptake of D-glucose derivatives in human U87 cells (200,000 cells/well) after incubation for 1 h with compounds **15–17** (50–100 μM) in Krebs–Ringer buffer in the presence or not of the GLUT inhibitor cytochalasin B (20 μM) ($\lambda_{exc/em}$ 488/610 nm (**15**), 635/670 nm (**16**), 355/450 nm (**17**)). Phototoxicity in U87 cells after incubation with compound **15** (100 μM) in the presence or absence of cytochalasin B (20 μM). Cells were illuminated (10 mW, 37 J cm$^{-2}$) and viability was assessed 16 h post illumination using a TACSR MTT cell proliferation assay with values normalized to those in cells without cytochalasin B. Data presented as mean values ± SEM ($n = 3$ independent experiments). **d** Excitation wavelengths used for single-photon illumination (*corresponds to two-photon illumination) and singlet oxygen generation quantum yields determined using DPBF in EtOH (Note: DPBF was found insoluble in water) using Rose Bengal as a reference (68%)[61]. †Singlet oxygen generation quantum yield reported for TPE[62]. ¥Reported values for protoporphyrin IX, the main photosensitizer produced upon metabolism of 5-aminolevulinic acid (**5-ALA**)[63]. *P* values were obtained from two-tailed unpaired *t* tests. Source data are available.

treatment of compound **15** and light-induced cell death in most U87 cells—as shown by the lack of fluorescence in the E2Crimson channel—but not in HCN-2 cells, which are less glycolytic and metabolically-active (Fig. 5b). Plot profile analysis confirmed that HCN-2 cells remained intact after the treatment whereas U87 cells were devoid of E2Crimson fluorescence (Fig. 5c). Furthermore, we confirmed that cocultures which had not been treated with compound **15** or not exposed to light showed no signs of death in any cell type (Supplementary Fig. 6). Finally, we analyzed the solubility and phototoxicity profiles of compound **15** and 5-aminolevulinic acid (**5-ALA**, Fig. 5d), the current gold standard in image-guided surgery for brain tumor resection. **5-ALA** stimulates the production of protoporphyrin IX in tumor cells and is currently evaluated in clinical trials for postoperative and interstitial PDT[50]. Both compound **15** and **5-ALA** showed full water solubility in the high micromolar range, and we did not detect the formation of any aggregates with diameter size >10 nm. We also compared the in vitro toxicity of different concentrations of compound **15** and **5-ALA** in U87 glioblastoma cells and observed

that compound **15** was able to induce significant levels of cell death at similar doses as **5-ALA** and with complete lack of toxicity in HCN-2 cells (Fig. 5e). Taken together, these results highlight the biocompatibility and potential utility of compound **15** for light-controlled ablation of cancer cells with comparable phototoxicity to PDT agents under clinical evaluation.

**Compound 15 enables safe removal of microtumors in vivo.** Encouraged by the selective photoreactivity of compound **15** in metabolically-active cancer cells, we evaluated whether it could be used to eliminate small tumors derived from human glioblastoma cells in vivo. We used a zebrafish model where low numbers of glioblastoma cells (around 50 ± 10 cells per organism) were injected into the yolk sac of 2-day-old embryo and allowed to form microtumors[51–53]. In order to quantify the viability of tumor cells by fluorescence confocal microscopy, we used the above described U87-nlsCrimson cells since they express a live-cell fluorescent reporter that can be detected in the optically-transparent zebrafish. As expected, the injection of U87-nlsCrimson cells into zebrafish

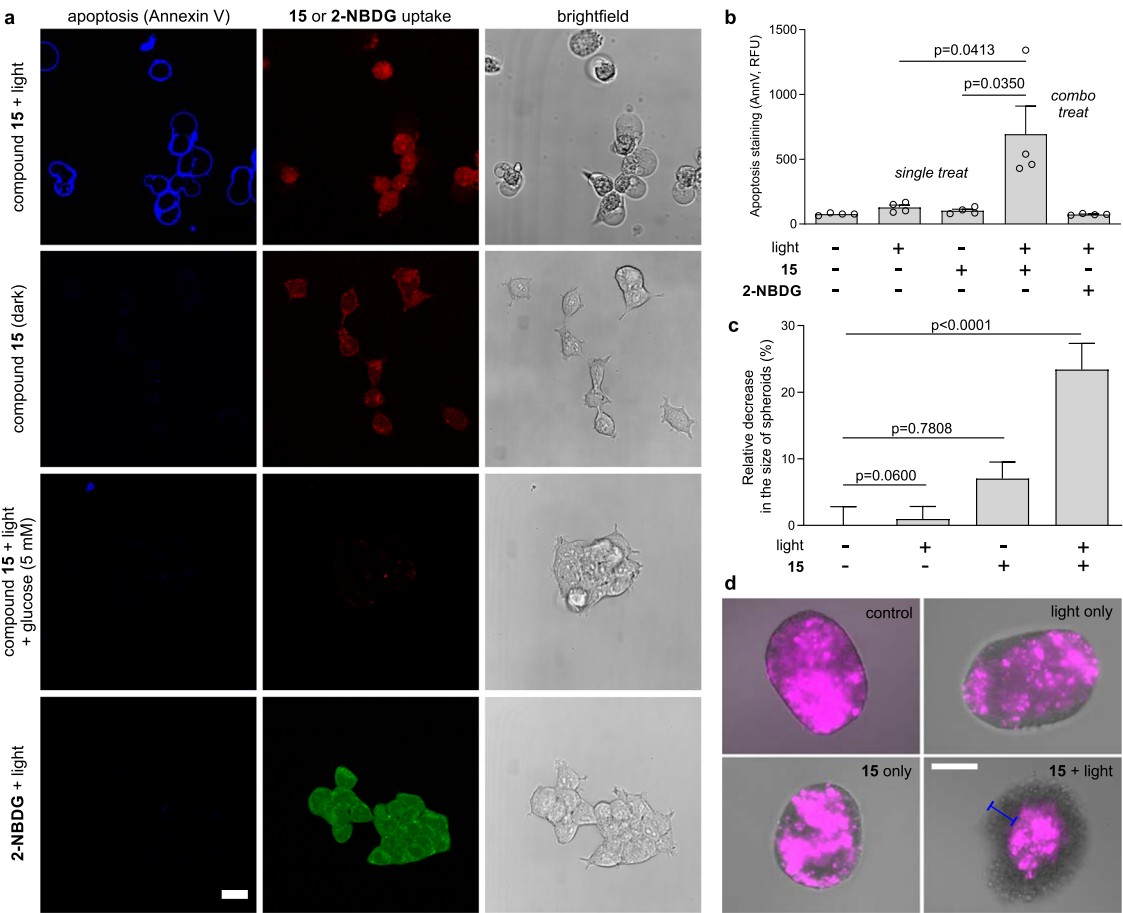

**Fig. 4 Compound 15 kills metabolically-active human glioblastoma cells in vitro and in 3D structures. a** Representative brightfield and fluorescence images (from three independent experiments) of U87 human glioblastoma cells (20,000 cells/well) treated with compound **15** (red, $\lambda_{em}$ 590 nm) or **2-NBDG** (green, $\lambda_{em}$ 530 nm) (both 100 μM) with or without D-glucose (5 mM) and illuminated or not with ThorLabs M530L3 LED (10 mW, 37 J cm$^{-2}$). Annexin V-Pacific blue (4 μg mL$^{-1}$) was used as an apoptosis marker (blue, $\lambda_{em}$ 450 nm). Scale bar: 10 μm. **b** Flow cytometric quantification (gating: Supplementary Fig. 5) of apoptotic U87 cells after different treatments. Data presented as mean values ± SD ($n = 4$ independent experiments). **c** Size decrease of U87-nlsCrimson 3D spheroids after variable treatments with compound **15** (100 μM) and visible light (10 mW, 37 J cm$^{-2}$). Data presented as mean values ± SEM and normalized to the viability of untreated spheroids ($n = 3$ independent experiments with three technical replicates). **d** Representative merged brightfield and fluorescence microscope images (from three independent experiments) of U87-nlsCrimson spheroids where live cells expressed E2Crimson fluorescent protein ($\lambda_{em}$ 645 nm, magenta). Spheroids treated with compound **15** plus light showed circumference of dead cells of ~50 μm in diameter (blue bar). Scale bar: 100 μm. *P* values were obtained from two-tailed unpaired *t* tests. Source data are available.

embryo at 48 hpf (hours post-fertilization) led to the formation of fluorescent tumor mass of several microns in diameter (Fig. 6a). Tumor-bearing zebrafish were then treated with compound **15** (microinjection of 6 pg into the yolk sac) and illuminated with visible light at a dose of 37 J cm$^{-2}$. Confocal microscopy images were taken from each group, and we quantified the number of remaining fluorescent U87 cells in each larva (Fig. 6a, b). The sequential treatment of compound **15** followed by light irradiation led to a dramatic reduction of the size of the microtumors, with some zebrafish being entirely free of glioblastoma cells after the treatment (Fig. 6b). The injection of compound **15** or the light irradiation of zebrafish on their own did not cause any significant reduction in the total number of cancer cells (Fig. 6b). Finally, we examined whether the in vivo treatment with compound **15** caused side effects in zebrafish. First, we compared the morphology of treated and untreated zebrafish, with no evident abnormalities between the two groups (Supplementary Fig. 7). Next, we examined their behavior and vital signs after the PDT treatment. Larvae from control and treated groups showed similar swimming behavior, as shown by the tracks of individual larva ($n \geq 10$ per group, Fig. 6c) with no significant differences in distance covered or velocity

(Fig. 6d and Supplementary Movies 1, 2). Heartbeat monitoring in both groups also revealed that treatment with compound **15** and light was safe and did not cause any evident deleterious side effects (Supplementary Movies 3 and 4). In summary, our data indicate that photosensitive metabolites represent a viable chemical strategy to harness the aberrant early metabolic features of aggressive cancer cells and to open avenues for minimally-invasive therapeutic strategies to safely eliminate microtumors.

In summary, this study demonstrates the utility of amino-substituted benzoselenadiazoles as small photosensitizers to kill metabolically-active cells in vitro and in vivo under full chemical and light control. Unlike in other current photosensitizers, the reduced size of benzoselenadiazoles retains key recognition properties of native metabolites and enables cell-selective uptake via various routes (e.g., peptidoglycan incorporation in bacteria, glucose transport in glioblastoma cells). This chemical approach opens therapeutic avenues for the selective ablation of cells that do not present discriminatory cell-surface markers, such as malignant cancer cells at early stages of disease, or that may become resistant to existing treatments, such as pathogenic bacteria. We also show that the dosing regime and light exposure

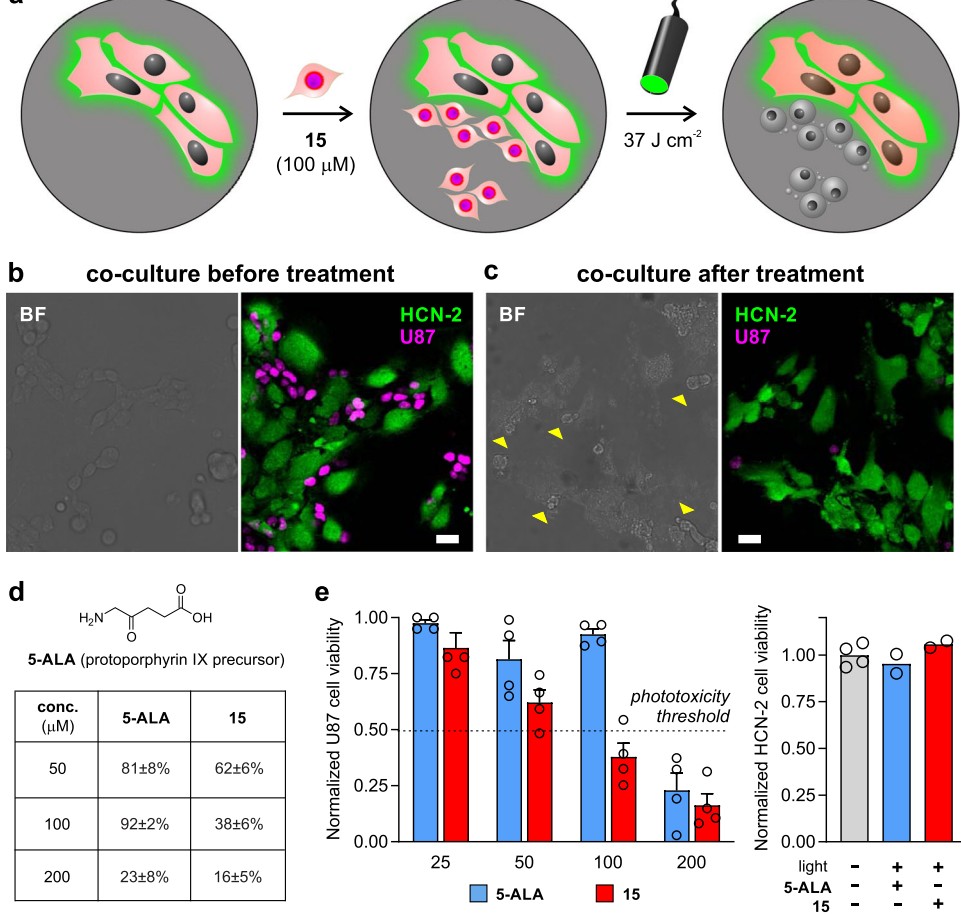

**Fig. 5 Compound 15 selectively ablates human glioblastoma cancer cells under clinical PDT conditions and in the presence of non-cancerous human brain cells. a** Procedure for coculturing human non-cancerous brain HCN-2 cells (stained with CellMask Green) and human glioblastoma U87-nlsCrimson cells. Cocultures (15,000 HCN-2 cells and 10,000 U87 cells per well) were incubated with compound **15** (100 μM) and illuminated with ThorLabs M530L3 LED (37 J cm$^{-2}$). The fluorescence emission from E2Crimson is reduced in U87-nlsCrimson glioblastoma cells upon cell death. (**b**, **c**) Representative brightfield and fluorescent confocal microscopy images (from three independent experiments) of cocultured HCN-2 (green, $\lambda_{em}$: 520 nm) and U87-nlsCrimson (magenta, $\lambda_{em}$: 645 nm) before (**b**) and after treatment (**c**). Yellow arrowheads in (**c**) highlight dead U87h cells devoid of Crimson fluorescence. Scale bars: 10 μm. **d** Phototoxicity analysis under different concentrations of compound **15** and **5-ALA** in U87 glioblastoma cells (20,000 cells/well) using the same irradiation settings (10 mW, 37 J cm$^{-2}$). Data presented as mean values ± SEM ($n = 3$ independent experiments). **e** HCN-2 cell viability (10,000 cells/well) after incubation with concentrations under the phototoxicity threshold [i.e., compound **15** (100 μM, red) and **5-ALA** (200 μM, blue)] and light exposure (37 J cm$^{-2}$). Data presented as mean values ± SEM ($n = 4$ independent experiments). Source data are available.

required for photosensitive metabolites meet high safety standards and are compatible with current procedures used in clinical PDT. The possibility to ablate cells with high precision and purely because of differential or altered metabolism will enable mechanistic studies to identify unknown subsets of disease-causing cells and to develop safer therapeutic approaches with enhanced spatiotemporal resolution.

## Discussion

The possibility to directly harness aberrant metabolism represents a strategy to improve the selectivity of PDT agents[54]. Photosensitizers that can enter target cells via endogenous metabolic transporters, especially those that are expressed differentially between target cells and their surrounding tissue, would increase selective cell ablation and minimize potential side effects. However, among the many chemical structures reported to date, very few photosensitizers can utilize endogenous metabolic transporters. Herein we describe benzoselenadiazoles as a generic photosensitizer scaffold to produce phototoxic metabolites. The synthesis of aminobenzoselenadiazoles

from small metabolites is straightforward, with high-yielding and scalable reactions (Fig. 1). Given that many metabolites contain reactive amino groups (e.g., amino acids, aminoglycans, peptides), this chemical strategy is translatable to a broad range of biological transporters and cell types.

The lack of charged chemical groups and the small size of benzoselenadiazoles are essential to retain recognition and uptake through cell-surface transporters. Aminobenzoselenadiazoles mimic the stereospecificity of chiral amino acids (as shown in L-Ala vs D-Ala derivatives, Fig. 2) and replicate the uptake of natural biomolecules, as demonstrated with the glucose analog **15** in GLUT transporters (Fig. 3). Of note, GLUT-mediated transport was not observed for other glucose-based photosensitizers, such as Nile Blue or TPE. We have demonstrated that the active uptake of amino-benzoselenadiazoles is essential to 1) minimize off-target toxicity in healthy cells, as shown in human cocultures and in vivo (Figs. 5 and 6), and 2) accumulate concentrations of photosensitizer inside target cells that can generate cytotoxic levels of singlet oxygen under clinical PDT exposure (Figs. 4, 5, and 6). Furthermore, given that aminobenzoselenadiazoles can be excited by single and two-photon

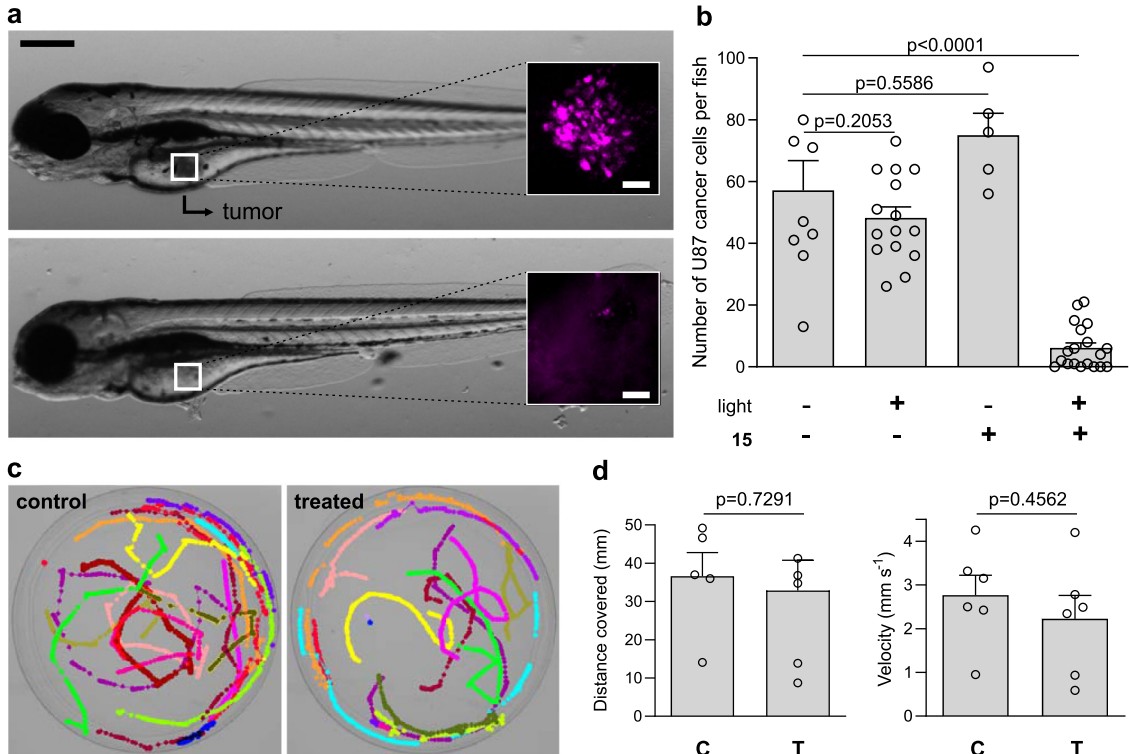

**Fig. 6 In vivo treatment of zebrafish xenografts with compound 15 followed by light irradiation leads to safe elimination of microtumors with no evident side effects. a** Representative brightfield microscopy images (from five independent experiments) of zebrafish larvae grafted with U87-nlsCrimson cells before treatment (top) and after the injection of compound **15** (6 pg per embryo) and light irradiation (37 J cm$^{-2}$, bottom). Insets show fluorescence microscopy images ($\lambda_{exc}/_{em}$: 561/645 nm) of microtumors formed by U87-nlsCrimson cells, which are fully eliminated after treatment. Scale bars: 50 μm (brightfield), 20 μm (fluorescence). **b** Quantification of fluorescent U87-nlsCrimson cells remaining in zebrafish larvae under different treatments. Data presented as means ± SEM ($n = 5$ independent experiments). **c** Microscope images of superimposed pseudo-colored swimming tracks of control untreated zebrafish larvae and double-treated (compound **15** plus light) larvae ($n = 5$ per group, representative videos for each group shown as Supplementary Movies 1 and 2). **d** Quantification of distance and velocity of control untreated zebrafish larvae (**C**) and double-treated zebrafish larvae (i.e., compound **15** plus light, **T**) larvae. Data presented as mean values ± SEM ($n = 5$ independent experiments). *P* values were obtained from one-way ANOVA with multiple comparisons in (**b**) and two-tailed unpaired *t* tests in (**d**). Source data are available.

sources (Fig. 3 and Supplementary Fig. 4), near-infrared multi-photon endoscopes—increasingly used for in vivo applications[55, 56]—could be considered as excitation sources. Recent in human studies combining local administration of benzodiazoles with endoscope-based imaging suggest that this approach might be translatable to the clinic[57].

Importantly, aminobenzoselenadiazoles show PDT efficacy at a power of 37 J cm$^{-2}$, which is compatible with clinical use[58]. Under these conditions, we have demonstrated that benzoselenadiazole D-amino acids can selectively kill bacterial cells upon irradiation. In mammalian cells, we observed similar cytotoxicity between compound **15** and **5-ALA**, the only FDA-approved photosensitizer able to enter cancer cells through endogenous transporters (i.e., PepT1 and PAT1 transporters)[59]. Unlike **5-ALA**, compound **15** enters cancer cells within minutes and does not need to be metabolized to be phototoxic (**5-ALA** is converted to PPIX after several hours). This feature allows local administration of the photosensitizer to minimize systemic side effects, something that cannot be done with **5-ALA**. In summary, this chemical platform offers a safe and rapid therapeutic avenue for the metabolic ablation of pathogenic cells, such as early-stage cancers and drug-resistant bacteria, which prove difficult to target with existing PDT approaches.

## Methods
**Density Functional Theory (DFT) calculations**. B3LYP, PBE0, M06-2X, and wB97XD functionals along with 6–31 G* and def2tzvp basis sets were tested for the closest match between the experimental absorption maxima in ethanol and the energy calculated for the $S^0 \rightarrow S^1$ transitions of compounds **2a–5a**. TD-B3LYP/def2tzvp achieved the closest match and was used for all subsequent computations.

**Bacterial cell growth**. *E. coli*, *B. subtilis*, and *S. aureus* cells were grown to exponential phase in LB medium at 37 °C. Cells were incubated with compounds **12**, **13**, or **NADA** at the indicated concentrations. Sacculi from labeled cells were isolated by adding cell suspensions to boiling sodium dodecyl sulfate (SDS, 5% w/v) and incubation for 30 min[41]. The SDS-insoluble material was collected by ultra-centrifugation at 39,000x*g* for 10 min at 30 °C, resuspended in water and boiled again in SDS (4% w/v) with stirring for 30 min. Samples were then washed, pelleted, and resuspended in the minimal amount of water. Growth curves were measured via absorbance at 600 nm over 18 h post-incubation with probes and/or illumination. Time-lapse microscopy experiments were performed with pre-labeled *E. coli* cells spotted on LB-agarose pads with images acquired every 1 min using a Nikon Ti2 Eclipse inverted microscope equipped with a Lumencor SPECTRA light engine and consecutively irradiating cells under optimized conditions for **NADA** ($\lambda_{exc}/_{em}$: 470/530 nm, 196 mW) and compound **13** ($\lambda_{exc}/_{em}$: 508/590 nm, 62 mW).

**Cell culture**. U87 cells (ATCC), U87-nlsCrimson cells (provided by Dr Dirk Sieger (University of Edinburgh)), and HCN-2 cells (ATCC) were grown in DMEM medium supplemented with 10–20% FBS, antibiotics (100 U mL$^{-1}$ penicillin, 100 mg mL$^{-1}$ streptomycin), and 2 mM L-glutamine at 37 °C in a humidified atmosphere with 5% CO$_2$. Cells were passaged in T-75 cell culture flasks using trypsin-EDTA.

**Photoviability experiments**. U87 cells were incubated in Krebs–Ringer bicarbonate buffer (KRB) for 2 h, incubated with or without compounds (**2a–17**) at the indicated concentrations and illuminated or not with either a ThorLabs M530L3/M625L4 LED or a white light source (10 mW, 37 J cm$^{-2}$). Medium was replaced with fresh DMEM, and cell viability was measured after 24 h using an MTT Cell

Proliferation assay following the manufacturer's instructions. Values were normalized to the viability of untreated cells and analyzed by GraphPad Prism.

**Determination of singlet oxygen quantum yields**. Solutions of DPBF (300 μM) and compounds **2a**–**16** (50 μM, unless indicated otherwise) in EtOH were illuminated with visible light (0.3 mW) for 0.5–4 min, followed by absorbance measurements at 410 nm using a Synergy spectrophotometer. Rates of singlet oxygen production (k) were calculated by plotting change in absorption of DPBF at 410 nm against time using the equation below.

$$k = \left( \left( \frac{\ln(\text{DPBF Abs}_t)}{\ln(\text{DPBF Abs}_0)} \right) \ln(\text{PSAbs}_{exc}) \right)$$

Relative singlet oxygen quantum yields were determined using the equation below with Rose Bengal as the reference. k: rates of singlet oxygen production, $\Phi_\Delta$: singlet oxygen quantum yields.

$$\Phi_{\Delta PS} = \Phi_{\Delta ref} \frac{k_{PS}}{k_{ref}}$$

**Flow cytometry**. U87 cells (300,000 cells/well) were incubated in glucose-free KRB buffer including or not compound **15** (100 μM) for 1 h. Cells were illuminated or not with visible light (10 mW). Following incubation in fresh DMEM for 16 h, cells were detached, resuspended in KRB buffer, and analyzed by flow cytometry using Annexin V-Pacific Blue™ (4 μg mL$^{-1}$, $\lambda_{exc/em}$: 405/450 nm). Flow cytometry data were analyzed using FlowJo.

**Live-cell fluorescence confocal microscopy**. U87 cells were plated in glass chamber slides (Nunc™ Lab-Tek™ II) in supplemented DMEM medium (20,000 cells/well). After 4 h, cells were incubated with or without compounds **15** or **2-NBDG** (100 μM) in KRB buffer for 1 h and illuminated or not with visible light (10 mW, 37 J cm$^{-2}$). Prior to imaging experiments, cells were incubated with Annexin V-Pacific Blue™ (4 μg mL$^{-1}$) and imaged in a Leica SP8 confocal microscope ($\lambda_{exc/em}$: 405/450 nm for Annexin V, $\lambda_{exc/em}$: 488/530 nm for **2-NBDG**, and $\lambda_{exc/em}$: 488/590 nm for compound **15**). All images were processed and analyzed using ImageJ.

**Glioblastoma spheroids**. U87-nlsCrimson cells were plated in Corning ultra-low attachment 96-well plates (500 cells/well) in supplemented DMEM medium and incubated for 72 h at 37 °C in a humidified atmosphere with 5% CO2. Spheroids were incubated with or without compound **15** (100 μM) for 1 h and illuminated or not with visible light (10 mW, 37 J cm$^{-2}$) before being washed and imaged on a widefield fluorescence EVOS microscope (Olympus) ($\lambda_{exc/em}$: 610/645 nm). The spheroids diameter was measured after 48 h incubation at 37 °C and size decrease measurements were normalized to untreated spheroids. All images were processed and analyzed using ImageJ.

**Human brain cell cocultures**. HCN-2 cells (15,000 cells/well) were plated in 6-well glass chamber slides (Nunc™ Lab-Tek™ II) and incubated with CellMask Green for 30 min. U87-nlsCrimson cells (10,000 cells/well) were then added to the wells and co-incubated for 4 h at 37 °C in a humidified atmosphere with 5% CO2. Cocultures in glucose-free KRB buffer were incubated with or without compound **15** for 1 h and illuminated or not with visible light, followed by imaging in Leica SP8 confocal microscope ($\lambda_{exc/em}$: 488/520 nm for CellMask Green and $\lambda_{exc/em}$: 561/645 nm for E2Crimson). All images were processed and analyzed using ImageJ.

**In vivo zebrafish experiments**. Experiments were carried out in accordance with Committee-approved ethical procedures. 48 hpf larvae were anesthetized using MS222 and mounted in 3% methylcellulose. U87-nlsCrimson (~50 cells/fish) were grafted into the edge of the yolk sac of larvae using a glass needle. Successfully-grafted larvae were selected under a dissecting fluorescence microscope. Larvae were injected with compound **15** (6 pg/embryo) and illuminated or not with visible light (10 mW, 37 J cm$^{-2}$). The tumor mass in grafted larvae was imaged at 24 h posttreatment by embedding in low-melting agarose and acquiring fluorescence images of U87-nlsCrimson cells in a Leica SP8 confocal microscope ($\lambda_{exc/em}$: 561/645 nm). Images were processed and analyzed using Imaris. Zebrafish were handled and maintained in the BVS aquatic facility at the University of Edinburgh according to standard protocols[60]. Animal experimentation was approved by the ethical review committee of the University of Edinburgh and the Home Office in accordance with the Scientific Procedure Act 1986.

**Reporting summary**. Further information on research design is available in the Nature Research Reporting Summary linked to this article.

## Data availability
The source data underlying Figs. 1b, 2c, 3c, 4b, c, 5e, 6b, d and Supplementary Figs. 2 and 4 are provided as a Source data file. Additional data that support findings of this study are available from the corresponding authors upon reasonable request. Source data are provided with this paper.

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

## Acknowledgements

The authors acknowledge funding from Medical Research Scotland (S.B.: 879-2015), MSCA Individual Fellowship (A.F.: 704912), Wellcome Trust Sir Henry Dale Fellowship (Y.F.: 100104/Z/12/Z), Cancer Research UK Early Detection Award (Y.F.: C38363/A26931), the Medical Research Council (A.M.E.: MR/N013166/1) and the National Research Foundation by the Ministry of Science, ICT & Future Planning, South Korea (J-S.L.: NRF-2018M3A9H4079286). M.V. acknowledges funds from ERC Consolidator Grant (771443). The authors thank the technical support from the Flow Cytometry, the Confocal Advanced Light Microscopy, and the BVS aquatic facility units at the University of Edinburgh.

## Author contributions

S.B., F.d.M., and A.F performed chemical synthesis and characterization of the compounds. J-.S.L. carried out DFT calculations. E.K., N.L.A., and J-.S.L. designed and performed bacterial experiments. S.B. designed and performed experiments in eukaryotic cells. L.H. and D.S. provided cell lines. I.R.B., A.M.E., and Y.F. performed zebrafish experiments. S.B. and M.V. wrote the manuscript with contributions from all authors. M.V. supervised the overall project.

## Competing interests

The University of Edinburgh has submitted the patent application "Small molecule photosensitizers for photodynamic therapy" (ID: WO2020/187913) covering some of the work described in this manuscript, where S.B., F.d.M, A.F., and M.V. are named as inventors. The remaining authors declare no competing interests.
