## [Peer Review File · Nature Communications]

REVIEWER COMMENTS

Reviewer #1 (Remarks to the Author):

The authors have substantially revised the manuscript. There are a few lingering issues:

1. DPBF is poorly soluble in water. The ROS generation detection in water should be investigated using SOSG as a probe.
2. In vitro cellular uptake and ROS generation with DCF-DA as a probe should be added.
3. The format of the references should be consistent and follow the journal style.

Reviewer #3 (Remarks to the Author):

The manuscript does not improve the novelty of this work, the claimed advantages are overemphasized in this manuscript. The characterizations are still insufficient and some major concerns are not resolved. This manuscript is recommended for a specialized journal after further in vitro and in vivo explorations.

Reviewer #4 (Remarks to the Author):

The authors have well addressed most of my previous comments and revised the manuscript accordingly. Regarding the comments about the fluorescence of compound 15 in spheroids and zebrafish (no. 5 and 6), the authors have also explained the technical difficulty. I think the revised manuscript in its present form is suitable for publication.

Response to reviewers (manuscript ID: NCOMMS-20-42795A-Z)

Reviewer 1

The authors have substantially revised the manuscript. There are a few lingering issues:

1. DPBF is poorly soluble in water. The ROS generation detection in water should be investigated using SOSG as a probe.

Answer: We have investigated the generation of ROS by compound **15** in water using both single-photon as well as multi-photon illumination and SOSG as a probe. The new results are included in the revised Figure S4 in the Supplementary Information.

2. In vitro cellular uptake and ROS generation with DCF-DA as a probe should be added.

Answer: We have used the DCFH-DA -one of the most widely used probes for detecting intracellular oxidative stress- to investigate the ROS generation in U87 cells following incubation with compound **15** and illumination. The new results indicate an increase in intracellular ROS upon treatment, and they are presented in the revised Figure S4 in the Supplementary Information.

3. The format of the references should be consistent and follow the journal style.

Answer: We have double-checked the format of all references in the revised manuscript.